# NTS, NTSR1 and ERs in the Pituitary–Gonadal Axis of Cycling and Postnatal Female Rats after BPA Treatment

**DOI:** 10.3390/ijms24087418

**Published:** 2023-04-18

**Authors:** Miriam González-Gómez, Ricardo Reyes, Mª del Carmen Damas-Hernández, Xiomara Plasencia-Cruz, Ibrahim González-Marrero, Rafael Alonso, Aixa R. Bello

**Affiliations:** 1Departamento de Ciencias Médicas Básicas, Área de Anatomía Humana, Facultad de Ciencias de la Salud, Universidad de La Laguna, 38200 La Laguna, Spain; xiomara-plasencia@hotmail.com (X.P.-C.);; 2Instituto de Tecnologías Biomédicas (ITB), 38200 La Laguna, Spain; 3Instituto Universitario de Neurociencia (IUNE), Universidad de La Laguna, 38200 La Laguna, Spain; 4Departamento de Bioquímica, Microbiología, Biología Celular y Genética, Área de Biología Celular, Facultad de Ciencias, Sección de Biología, Universidad de La Laguna, 38200 La Laguna, Spain; abello@ull.edu.es; 5Instituto de Enfermedades Tropicales y Salud Pública de Canarias (IUETSP), 38296 La Laguna, Spain; 6Departamento de Psicología Clínica, Psicobiología y Metodología, Universidad de La Laguna, 38200 La Laguna, Spain; mdamas@ull.edu.es; 7Departamento de Ciencias Médicas Básicas, Área de Fisiología, Facultad de Ciencias de la Salud, Universidad de La Laguna, 38200 La Laguna, Spain

**Keywords:** neurotensin, neurotensin receptor 1, estrogen receptors, bisphenol-A, pituitary gland, ovaries

## Abstract

The neuropeptide neurotensin (NTS) is involved in regulating the reproductive axis and is expressed at each level of this axis (hypothalamus–pituitary–gonads). This dependence on estrogen levels has been widely demonstrated in the hypothalamus and pituitary. We focused on confirming the relationship of NTS with estrogens and the gonadal axis, using a particularly important environmental estrogenic molecule, bisphenol-A (BPA). Based on the experimental models or in vitro cell studies, it has been shown that BPA can negatively affect reproductive function. We studied for the first time the action of an exogenous estrogenic substance on the expression of NTS and estrogen receptors in the pituitary-gonadal axis during prolonged in vivo exposure. The exposure to BPA at 0.5 and 2 mg/kg body weight per day during gestation and lactation was monitored through indirect immunohistochemical procedures applied to the pituitary and ovary sections. Our results demonstrate that BPA induces alterations in the reproductive axis of the offspring, mainly after the first postnatal week. The rat pups exposed to BPA exhibited accelerated sexual maturation to puberty. There was no effect on the number of rats born per litter, although the fewer primordial follicles suggest a shorter fertile life.

## 1. Introduction

Neuropeptides are peptides between 3 and 100 aa, that can act as neurotransmitters and neurohormones. It is well documented that estrogens augment the effects of neuropeptide expression on the neuroendocrine axes [1,2,3,4,5,6,7,8].

Neurotensin (NTS) is a neuroactive tridecapeptide that acts as a neurohormone and is present in the preoptic area and the pituitary gland in mammals. It is strongly enhanced by estrogen [1,2,3,4,5,6,7] through cAMP-mediated mechanisms [7]. NTS exerts its effects through two receptors, NTSR1 and NTSR2 [9], as well as the type Sortilin 1 (Sort1) receptors [10]. The NTSR1 was the first to be identified and referred to as “high affinity.” NTS is one of the main regulatory factors of pituitary secretory activity [11,12,13,14,15]. So far, its most studied function in the regulation of pituitary hormones concerns its modulation of the gonadal axis [4,16], however, information on the roles of neurotensin in reproduction is scarce. The action of NTS on the hypothalamic and pituitary axes is mediated mainly by NTSR1 [11]. Thus, NTS is widely expressed at the hypothalamic level, where it has a modulatory role in estrogen-mediated GnRH release [17,18].

Moreover, both NTS expression in hypothalamic neurons and its release into the portal blood are under estrogenic control [1,19]. In the pituitary, we previously reported that the gonadotrophs are also NTS-immunoreactive (NTS-ir) [6,20,21] and express the high-affinity receptor NTSR1 [21]. This suggests an autocrine function for NTS in these cells. We also showed that these NTS-ir pituitary cells are also ERα-ir and that the distribution pattern of NTS-ir cells in the pituitary gland partially depends on gonadal function [6,20,21,22,23]. Recent studies using experimental models have confirmed NTS expression in gonads [24,25]. These authors show that granulosa cells are likely the primary or only source of NTS in the ovulatory follicle, although it has also been put forward as a paracrine mediator of ovulation [25]. These data indicate that NTS is largely involved in the regulation of the hypothalamus–pituitary–gonadal axis (HPG) and that the estrogen environment plays a critical role in the normal expression of NTS and its distribution pattern throughout this axis. Alterations in estrogen levels can cause abnormal expression of NTS and consequently affect the normal function of the HPG axis.

Numerous substances present in the current-day human environment have been shown to bind to steroid and sex-hormone receptors and thus either mimic or inhibit their actions. This interference may result in cellular effects equivalent to hormone over- or underexposure and, as a consequence, disturb the endocrine system, so they have been termed “endocrine disrupters” [26,27,28]. In this respect, the development of estrogen target tissues appears to be particularly vulnerable to the effects during the prenatal period of some of these compounds, also known as xenoestrogens [29,30,31]. Given that rodents are especially responsive to estrogen in utero, it is not surprising that exposure to natural or synthetic estrogens during the prenatal period can have a wide variety of developmental effects [32,33].

Bisphenol-A (BPA) is a particularly important environmental estrogenic molecule [34], which has the potential to disturb the endocrine system in both wildlife and humans. BPA is a monomer of polycarbonate plastic used in diverse materials in contact with foodstuffs through its inclusion in plastics and epoxy resins. It can be eluted easily from these materials, which leads to its inclusion in industrial food production and thus the food chain, resulting in chronic exposure to this molecule [35]. BPA is a molecule composed of two unsaturated phenolic rings and was discovered as an estrogenic molecule in 1936 by Doods and Lawson [36]. In both in vitro and in vivo systems, mild estrogenic activity was indicated [37], binding to both nuclear estrogen receptors (ERα and ERβ) and membrane estrogen receptor GPR30 [38,39,40] with low affinity and causing reporter-gene transactivation in vitro [28,41]. Traditionally, BPA has been considered a weak estrogen, based on its lower binding affinity to the nuclear estrogen receptors (ERs) compared to 17β-estradiol (E2) as well as its low transcriptional activity after ER activation. Mounting evidence suggests a variety of pathways through which BPA can elicit cellular responses at very low concentrations with the same or even higher efficiency than E2. In addition, BPA can act through alternative mechanisms [42,43].

Oocyte, embryonic, and postnatal development are particularly sensitive times for exposure to chemicals, which can be most harmful in these periods. The offspring of pregnant rats exposed to BPA during the gestational period can display several altered reproductive functions, depending on dosage and experimental conditions [31,44,45,46], because it can cross the placenta and reach fetal organs [47,48,49]. Another vulnerability factor is the low capacity of the fetal and newborn liver to deactivate BPA [50]. In humans, BPA is commonly ingested from various sources [34,51,52], and a range of estrogenic actions have been reported [50,53,54], such as early sexual maturation in females, increased frequency of abnormal penile/urethral development in males, childhood and adult obesity, type-2 diabetes, and hormonally mediated neoplasms such as prostate and breast cancers (reviewed in [51,54,55,56]. In experimental animals, it causes uterotrophy [57], decreased sperm production [58,59], stimulation of prolactin release [60], and promotion of cell proliferation in a breast cancer cell line [37]. BPA also affects granulosa cell steroidogenesis in vitro [61].

The hormonal changes the female rat undergoes during the cycle involve important changes in the expression of NTS and its receptors [6], as well as in the presence of ERs [22]. The presence of NTS in ovarian, granulosa, and stromal cells has been demonstrated experimentally in humans and rats [24]. As we have previously shown, the expression of NTS in the gonadal axis is induced by endogenous estrogens [6,21], but the action of exogenous estrogens on this expression is unknown [62,63]. We have focused this work on the expression of nuclear ERs (ERα/β), NTS, and its receptor NTSR1 in the hypophysis-gonadal axis of female rat offspring and their mothers treated with BPA during gestation and lactation.

## 2. Results

### 2.1. NTS, NTSR1, and ERs in Ovaries of Cycling Adult Females

The immunohistochemical study of the ovary showed distributional differences in NTS-immunoreactive (NTS-ir) and NTSR1-immunoreactive (NTSR1) cells during the analyzed phases of diestrus and proestrus. In diestrus, NTS-ir was observed in oocytes, granulosa cells in the different follicles, and corpus luteum cells (Figure 1a,b). In proestrus, NTS-ir was detected exclusively in oocytes in the different follicles (Figure 1c). In diestrus, there was NTSR1-ir in the granulosa cells of the follicles, whereas in proestrus it was localized in the oocytes and theca cells of mature follicles (Figure 1d–f). The ERs study showed a predominance of ERβ versus ERα. Additionally, in diestrus, immunoreactivity for both receptors was observed in oocytes, granulosa, theca, and corpus luteum cells, whereas in proestrus, it was found only in oocytes (Figure 1g–l). In all cases, ERα-ir was cytosolic and nuclear, whereas ERβ-ir was exclusively cytosolic (Figure 1g–l).

### 2.2. General Effects of BPA Treatment on Female Offspring

No significant differences were observed in the number of offspring per litter between the control and BPA-treated groups (Figure 2A). However, BPA groups showed a significant reduction in ovarian weight at both the first, second, and fourth weeks of postnatal life (Figure 2B) and at the time of vaginal opening (Figure 2C).

#### Effects of BPA Treatment on Offspring Ovaries

A histological and histomorphometry study of the ovaries from the first week of postnatal life showed a significant reduction in the number of primordial and primary monolayer follicles. However, the number of primary multilayer and secondary follicles significantly increased in the BPA-treated groups compared to the control group (Figure 3A,B). In the second and third weeks of postnatal life, the reduction in the number of primordial follicles was maintained but less pronounced than in the first week (Figure 3B). No differences were observed in this period in the number of primary monolayer follicles between the BPA-treated and control groups (Figure 3B). During the third week, BPA did not modify the numbers of the other types of follicles (primary monolayer, secondary and antral) in some cases, while in others it led to slight increases or decreases versus the control group (Figure 3B).

### 2.3. Effects of BPA Treatment on NTS, NTSR1, and ERα/β Expression/Immunoreactivity in Female Offspring

#### 2.3.1. Pituitary Gland (NTS and NTSR1)

In the first week of postnatal life, sparse NTS-ir cells—which were also TSH-ir—were observed in the central area of the anterior pituitary lobe in both control and BPA-treated animals, at both doses of BPA (Figure 4a–e). Throughout the second and third weeks, there was a progressive increase in NTS-ir/TSH-ir cells in the central zone of the anterior lobe in all groups (Figure 4f–j).

At the end of the third week, NTS-ir cells—which were also FSH-ir—were detected at the margins of the anterior lobe in all groups (Figure 4k–o). There were NTSR1-ir cells in the central zone of the anterior lobe of all groups in the first week of postnatal life (Figure 5a,b). Throughout the second and third weeks, NTSR1-ir cells increased throughout the anterior lobe (Figure 5c,d).

In dams after weaning, when estrogen levels are high, NTS-ir cells were identified in both the central zone and the margins of the anterior lobe (Figure 6a,c,e), but only NTSR1-ir cells in the central zone of the anterior lobe (Figure 6b,d,f). No differences were observed between the control and BPA-treated groups in the distribution of NTS-ir and NTSR1-ir cells in either mothers or offspring.

#### 2.3.2. Pituitary Gland (ERα/β)

In the first week of postnatal life, ERα-ir cells were observed in the anterior lobe of the pituitary gland of control and BPA-treated animals at both doses of BPA. While in the control groups, ERα-ir cells were scarce, in the BPA-treated groups they were numerous, with the BPA-2 group showing the highest proportion and intensity of immunoreactivity (Figure 7a,c,e). From the second week on, the presence of ERα-ir cells was similar in all groups. At all times, ERα-ir was nuclear. In dams after weaning, ERα-ir cells were seen in the anterior lobe in both the control and BPA-treated groups. However, whereas in the control group, the ERα-ir was nuclear, in the BPA-treated groups it was cytosolic (Figure 7b,d,f). No immunoreactivity for ERβ was observed during the period analyzed in either control or BPA-treated animals in the pituitary gland.

#### 2.3.3. Ovaries (NTS and NTSR1)

In the first week of postnatal life, no NTS-ir cells were observed in the ovaries of control animals, while in the BPA-treated groups, NTS-ir oocytes were present in primordial and primary follicles (Figure 8a–c). In the second week, NTS-ir oocytes were detected in primordial follicles in the control group, whereas in the BPA-treated groups, there was NTS-ir in primordial and primary follicles, as well as in granulosa and theca cells of primary follicles (Figure 8d–f). In the third week, NTS-ir oocytes were found in primordial and primary follicles in both, the control and BPA-treated groups, as well as in granulosa and theca cells of primary and secondary follicles (Figure 8g–i).

In contrast, in the first postnatal week (PW), NTSR1-ir oocytes were found in the primordial follicles of control and BPA-treated animals, although with a lower intensity of immunoreactivity in the BPA-2 group (Figure 9a–c). Likewise, there were NTSR1-ir granulosa cells in some primary follicles of BPA-treated animals (Figure 9b,c). From the second week on, NTSR1-ir was maintained in the oocytes of primordial follicles, whereas NTSR1-ir granulosa cells were more numerous in primary follicles (Figure 9d–f).

In dams after weaning, NTS-ir and NTSR1-ir cells were observed in oocytes and granulosa cells in all groups (Figure 10a–f), with the only difference between the control and BPA-treated groups being the higher proportion of NTS and NTSR1-ir oocytes in BPA-treated animals.

#### 2.3.4. Ovaries (ERα/β)

No ERα-ir cells were found in the ovaries of control animals during the first three weeks of postnatal life. Contrastingly, in BPA-treated animals, there were ERα-ir granulosa cells in secondary follicles from the first week of postnatal life onward (Figure 11a,b). From the second week on, granulosa ERα-ir cells increased in secondary follicles (Figure 11c,d).

No ERβ-ir cells were detected in the ovaries of control animals during the first week of postnatal life, whereas in the BPA-treated groups, there were numerous ERβ-ir oocytes in primordial follicles (Figure 12a–c). From the second week on, ERβ-ir was present in oocytes from primordial and primary follicles, as well as in some granulosa cells, in both control and BPA-treated animals (Figure 12d–f). In the third week, all groups showed numerous ERβ-ir granulosa cells in primary and secondary follicles (Figure 12g–i).

In dams after weaning, the distribution was the same for both estrogen receptor types, α and β, in all three groups (control, BPA-0.5, and BPA-2), with immunoreactivity in oocytes and granulosa cells (Figure 13a–d).

## 3. Discussion

This study demonstrates that BPA induces morphohistological changes in the gonads of postnatal female offspring of rats treated with BPA during pregnancy and lactation. Furthermore, BPA clearly acts as an estrogen agonist in the expression of the NTS peptide and its high-affinity receptor NTSR1, along with the intracellular estrogen receptors α and β, in the pituitary-gonadal axis of the postnatal and cycling female rat, in vivo. However, results in the BPA-treated dams suggest an antagonistic effect on estrogen receptors in the pituitary gland. For the first time, our results show NTS-ir in the oocytes at different maturation stages of ovarian follicles.

No differences were found between the two doses used, and a non-monotonic response was detected, neither in the pituitary gland nor in the ovary. It must be considered that we have only used two doses and we do not know if an intermediate dose in the range of 0.5 to 2 mg, or even higher, would have produced changes in another way. In addition, in this work, we have only analyzed the response in females, and the response also differs according to sex. On the other hand, other types of studies would be necessary to find out if there have been more changes at the cellular level and if these changes are maintained between generations with the doses used.

In the literature, we found no data on the action of BPA on gonad weight among the effects on the offspring of pregnant rats exposed to this compound. Our data show a significant reduction in ovarian weight in the treated groups with respect to the control groups, at least in the first four PW. This drop may be due to the significant decrease in primordial follicles, mainly in the first week of life. Follicular development during fetal life, oocyte development, and their lodging in structures called primordial follicles occur in most mammals before or in the first postnatal days [64,65]. The estrogen dependence of primordial follicle formation is well documented in various mammalian species [66,67]. In all these species, although primordial follicles form at different times according to experimental procedures, a drop in estrogen level is necessary for the process to take place [68,69]. Studies have shown that low-level exposure to estradiol in mice induces cell assembly processes during primordial follicle formation, whereas high levels cause cell degeneration and prevent them from forming [35,65,69,70]. Our results suggest that estrogen levels increased after BPA treatment, causing a reduction in the number of primordial follicles, which demonstrates the estrogenic action of BPA.

The lower number of primordial follicles found in our study has been observed in both rats and other mammalian species after treatment with BPA. However, the experimental conditions, BPA doses, and treatment duration differed in these reports [71]. In addition, the morphological and histological analysis suggests that pre- and postnatal BPA exposure has an important effect on gonadal development by acting as an estrogen agonist. Estrogen action during development is crucial for normal gonadal development in general. It has been shown in some mammalian species that the absence of estrogens in early embryonic life affects normal ovarian development, as they are fundamental in the process of germ cell proliferation and the initiation of primordial follicle formation [68]. Therefore, disturbed estrogen exposure can have permanent adverse consequences, such as tumor formation [72,73]. The onset of sensitivity to endogenous or exogenous estrogens is clearly related to the ontogeny of ERα and/or ERβ expression, which varies in different organs, tissues, and cells. In previous work [22], we showed the predominance of ERα in pituitary cells and its changes concerning the timing of the rat estrous cycle. Furthermore, in these rat studies in vivo, we demonstrated how gonadal steroids induce the expression of the gonadal axis regulator neurotensin in hormone-producing pituitary cells [6] and also how, in postnatal development, NTS-ir cells are present from the first week [21].

In the present work, we found that the action of exogenous estrogen, BPA, did not change the pattern previously described for the ERs, although a greater number of immunoreactive cells were observed in the treated groups. There is evidence regarding the action mechanisms of BPA in relation to estrogenic activity. BPA can activate transcription by linking to ERα and β [74], as well as to the ERβ coactivator [75] and to GPR30, a non-classical ER that seems to play an important role in the deleterious effects of low doses of BPA [76]. Among such impacts of BPA on the ovulatory process, oocytes can become defective due to damage to various organelles [77], as well as increased levels of apoptosis under exposure to high doses [78]. At the organelle level, BPA can cause abnormal mitochondrial distribution, endoplasmic reticulum stress, abnormal structure of the Golgi apparatus, and lysosomal damage. These abnormal organelle functions lead to wider regional oocyte dysfunctions, which ultimately disrupt oocyte maturation [77].

Strikingly, our results on ERs expression indicate that the most intense impacts of BPA occur at the early stages of postnatal development (first PW). Thus, indirect exposure to BPA has a detectable impact on offspring at both tested dosages during these early stages. Nevertheless, the effects were transient since they disappeared in adulthood. One of these effects is the increased expression of ERα in pituitary hormone producing cells of rat offspring treated with both low and high doses. This result is consistent with an estrogenic action since high levels of estradiol produce an increase in ERα [79,80]. The increased immunoreactivity against ERα in offspring pituitary cells after BPA treatment is also consistent with an increase in ERα mRNA detected in the pituitary cells after BPA exposure [81]. However, the effect on the mother seems to be the opposite, inhibiting the activity of receptors by preventing their transfer to the nucleus.

In the ovary, both nuclear estrogen receptors [82] are known to be regulators of granulosa cell differentiation, follicle and oocyte growth, and ovulation. Several papers have already proposed roles for each of these receptors in ovarian physiology [83,84,85], with ERβ playing an important role in the follicle maturation process and ovulation [86]. These studies have used knockout models and gene expression techniques. In addition, these and later authors [87] assign the ERβ a role in gametogenesis through its expression in granulosa cells, while ERα is important in steroidogenesis and in relation to theca cells. In our work under in vivo physiological conditions, we found ERα and β-ir in granulosa cells, the corpus luteum, and the stroma of cycling rats in the diestrus phase. Additionally, ERα-ir was also evident in the nucleus and oocyte cytoplasm in follicles at several developmental stages, and ERβ-ir only in the oocyte cytoplasm in primordial and primary follicles of the cycling female rat in proestrus. The presence of receptors within the oocyte suggests a paracrine role for estrogens in its maturation process or an inducing role for other directly involved factors, such as neurotensin. Documentation on the role of this peptide in the ovary is very scarce, and only recently has its expression during the periovulatory period been described in human and rat ovaries with hCG hormone-induced expression [24].

Campbell et al. [25] show that NTS mRNA levels in granulosa cells increase after the ovulatory surge of gonadotropin and that granulosa cells appear to be the sole source of NTS. These authors confirm the role of NTS in the ovulatory process since NTS neutralization disrupts its progress; however, they did not probe the effects of NTS on the oocyte itself.

Exposure to BPA during pregnancy also alters oocyte development in unborn female fetuses and when they reach adulthood, the disturbances increase the number of chromosomally abnormal oocytes and embryos.

In the control rats, during postnatal development, ovarian tissue immunoreactivity to ERβ was detected before ERα, confirming the importance of this receptor subtype there. In this way, we detected ovarian ERβ-ir cells in control rats at the second PW, whereas other authors first detected them at postnatal day 5 (PND 5) in mice and rats [88,89]. Ovarian ERα-ir cells were observed in the control group from the fourth PW onward, although other authors report them from embryonic day 14 in mice and neonatal rats [88,89].

Consequently, ovarian NTS expression during postnatal development could be mediated by that of ERβ, which was detected at the second PW in the control group’s oocytes and several ovarian stromal cells. This suggests that NTS may have a role in the follicle and oocyte maturation as well as in estrogen production during this period. Furthermore, coinciding with the estradiol surge at the end of the second PW [90], we detected NTS-ir in numerous granulosa cells. At this stage, estrogenic action on NTS expression could be mediated by ERβ. During ovarian and pituitary postnatal development, we have shown that NTSR-1-ir appears before NTS-ir, as previously described for the hypothalamus [91,92,93,94]. Likewise, BPA treatment accelerates the expression of NTSR1 in the ovaries in comparison to controls, always before NTS.

We suggest that the experimental conditions used here have affected the formation of these follicles at a critical period, either prenatally or at birth. Furthermore, the fertile period of these offspring could be expected to be shorter since they have a poor reserve of primordial follicles. On the other hand, oogenesis is a complex process initiated during fetal development, with these first events in the fetal ovary also being a response to normal estrogenic action [95]. Furthermore, BPA may influence the first stages of meiosis, which are estrogen-sensitive [96].

Thus, it has been proposed that even low BPA exposure during gestation has multigenerational consequences, increasing the probability of producing abnormal chromosomes in the second generation [95]. According to these authors, BPA exposure during pregnancy disturbs oocyte development in unborn female fetuses and when they reach adulthood. Such disturbances result in more numerous chromosomally abnormal oocytes and embryos. Results from other groups have also shown that high BPA doses cause permanent changes when female reproductive organs are exposed neonatally, such as an irregular estrous cycle, a lack of corpus luteum, and a high number of cystic follicles [96,97,98]. In our experimental conditions, high and low doses of BPA could affect fertility at least in the offspring, considering the lower number of primordial follicles. Further studies would be necessary to understand and foresee other consequences of such litter. Overexpression of ERs induced by BPA would support its reported estradiol agonist action [79,80]. The increase in ERα could also induce NTS expression in the pituitary cells.

Our results show that exposure to BPA stimulates ER expression, with special relevance to the physiological effects, increasing responsiveness to this xenoestrogen as well as to the endogenous ligand. It could lead to wider organizational effects since the compensatory mechanisms (negative feedback) in the prenatal period have not yet been established. Nevertheless, direct exposure in the dams did not affect ER expression in either the pituitary or the ovary, probably because BPA acts as a reversible regulatory agent in adulthood. In prenatal development, however, it acts as an organizational agent. These results highlight the power of xenoestrogen exposure during perinatal development. Precocious development of gonads and the early vaginal opening in female offspring treated with BPA are signs of accelerated puberty onset.

## 4. Materials and Methods

### 4.1. Animals Experiments and Exposure to BPA

Sprague–Dawley rats were caged under a 14:10 light:dark cycle to mimic summer conditions in an air-conditioned room (23 ± 1 °C) with *ad libitum* access to standard rat food and tap water. They were left undisturbed for at least a week before the experiments began. All animal experiments were carried out in conformity with the European Directive (2010/63UE) on Care and Use in Experimental Procedures. Furthermore, the animal protocols were approved on 31 March 2023 by the Ethics Committee for Animal Care at the University of La Laguna (CEIBA) with the identification code CEIBA2023-3289. All experimental procedures were made under aseptic conditions.

Sexually mature female and male rats were mated. The morning on which a vaginal plug and sperm were observed was designated as day 1 of pregnancy. Fifteen pregnant rats were divided into three groups (five animals per group): vehicle-treated (control), 0.5 mg/kg-BW/day BPA (BPA-0.5), and 2 mg/kg-BW/day BPA (BPA-2). We have chosen a BPA dose of 0.5 mg/kg-BW per day, here considered low, because it is comparable to the minimum dose with estrogenic effect according to Markey et al. [99]. Within the possible range of doses used in this paper, we have chosen 2 mg/kg-BW per day as the high dose. Each rat received a subcutaneous injection of either BPA (CAS 80-05-7; lot 03105ES; purity ≥99%; Sigma Aldrich, St. Louis, MO, USA) suspended in sesame oil as the vehicle or sesame oil alone (control), every morning from gestation day 1 to the day before weaning (21 days after delivery). Each treatment group included 8–10 pregnant rats. The day of birth was designated postnatal day 0 (PND 0). After delivery, rats with offspring were housed in cages, and litter sizes were adjusted to 8–10 pups/dam at PND 5.

The number of rats born alive per litter was checked at PND 1. Five female animals from each dam per group were anesthetized with a dose of sodium pentobarbital of 200 mg/mL and sacrificed weekly from the first to the sixth postnatal week (PW), and the ovaries were weighed each time. Ovaries were embedded in paraffin, and sections were stained with hematoxylin and eosin for histological analysis. We used 6–8 female rats per group in each age and analyzed 6–10 sections of 20 µm per ovary using the same levels. The number of follicles per section was quantified and statistically analyzed. All female offspring were checked daily for the vaginal opening. From 50 to 90 days, estrous cyclicity was examined by vaginal smears in selected female rats (5 or 6 females per group).

The body weights of dams were checked once daily during the pregnancy period. All were terminally anesthetized and fixed by intracardiac perfusion of paraformaldehyde at the weaning of the offspring (PND 21). Six adult, non-pregnant female rats’ ovaries were processed for an immunohistochemical study regarding NTS, NTSR1, ERα, and β levels.

### 4.2. Antibodies

For ERα and ERβ, a commercial rabbit polyclonal IgG anti-ERα serum (H184, Santa Cruz Biotechnology Inc., Santa Cruz, CA, USA) directed against the N-terminus of human ERα was used, along with a commercial goat polyclonal IgG anti-ERβ serum (Y-19, Santa Cruz Biotechnology) against the N-terminus of mouse ERβ. Antisera for detecting NTS and its receptor (NTSR-1) were raised in rabbit by Dr. G. Tramu (University of Bordeaux, France), using the synthetic peptide and the sequence 375–387 of NTS receptor-1, respectively, as described elsewhere [21,100]. To identify the specific hormone-producing cells, polyclonal antisera provided by Dr. G. Tramu were used. These antisera against human βTSH and βFSH were developed in rabbits from synthetic antigens (Chemicon International, Los Angeles, CA, USA). Their immunological properties have been described elsewhere [101,102,103,104,105].

### 4.3. Immunohistochemistry

All rats were anesthetized with a dose of sodium pentobarbital of 200 mg/mL and fixed by intracardiac perfusion of sodium phosphate buffer (0.1 M pH 7.4) containing 4% paraformaldehyde and 0.2% picric acid. Pituitaries and ovaries were quickly dissected out and postfixed for an additional two hours in the same fixative and Bouin’s fixative, respectively. Pituitaries were immersed overnight in Tris-buffered saline (TBS; 0.05 M, pH 7.6) containing 20% sucrose, embedded in Tissue-Tek^®^, and then frozen in isopentane cooled with liquid nitrogen. Horizontal 8–10 µm cryostat sections were cut and collected on gelatin-coated slides. Ovaries were embedded in paraffin and then sectioned at 8 µm. After rehydration, sections were rinsed in TBS, which was used for all further incubations and washes.

The indirect immunohistochemical procedure started with incubating the sections overnight with ERα (1:250), ERβ (1:250), NTS (1:200), and NTSR-1 (1:300) antisera. After rinsing, peroxidase activity was revealed in 0.05 M Tris-HCl (pH 7.6) containing 0.004% of 4-chloro-1-naphthol (Sigma^®^, Welwyn Garden City, UK) and 0.003% H_2_O_2_. The specificity of immunostaining was assessed by replacing the specific antiserum with normal serum and omitting step one of the reaction. To demonstrate the colocation of NTS and hormones in the same pituitary cell, slides were treated as described above by using antisera to NTS and then processed according to the elution-restaining procedure [106], using antibodies against β-FSH (1/800) and β-TSH (1/800). Sections were incubated with peroxidase-conjugated goat anti-rabbit or anti-mouse IgG (Jackson Immunochemical, Baltimore, MD, USA). The sites of peroxidase attachment in the second immunoreaction were identified by incubation with 0.04% diethyl-carbazol and 0.01% H_2_O_2_ in acetate buffer (pH 5, 0.1 M).

### 4.4. Statistical Analysis

Data are expressed as mean ± SD. As the data were distributed normally and the variances were equal, a parametric one-way ANOVA followed by the “student *t*” was used to compare between groups. Differences between means are indicated at a level of *p* < 0.05.

## 5. Conclusions

In summary, our results confirm the estrogenic action of BPA at morphological and protein expression levels. The effects were more prominent in pups exposed during gestation and lactation, indicating that BPA can cross the placenta and be transmitted through maternal milk. This study only examined the first generation, so more research is needed to find out the long-term effects on subsequent generations and on tumor formation.

## Figures and Tables

**Figure 1 ijms-24-07418-f001:**
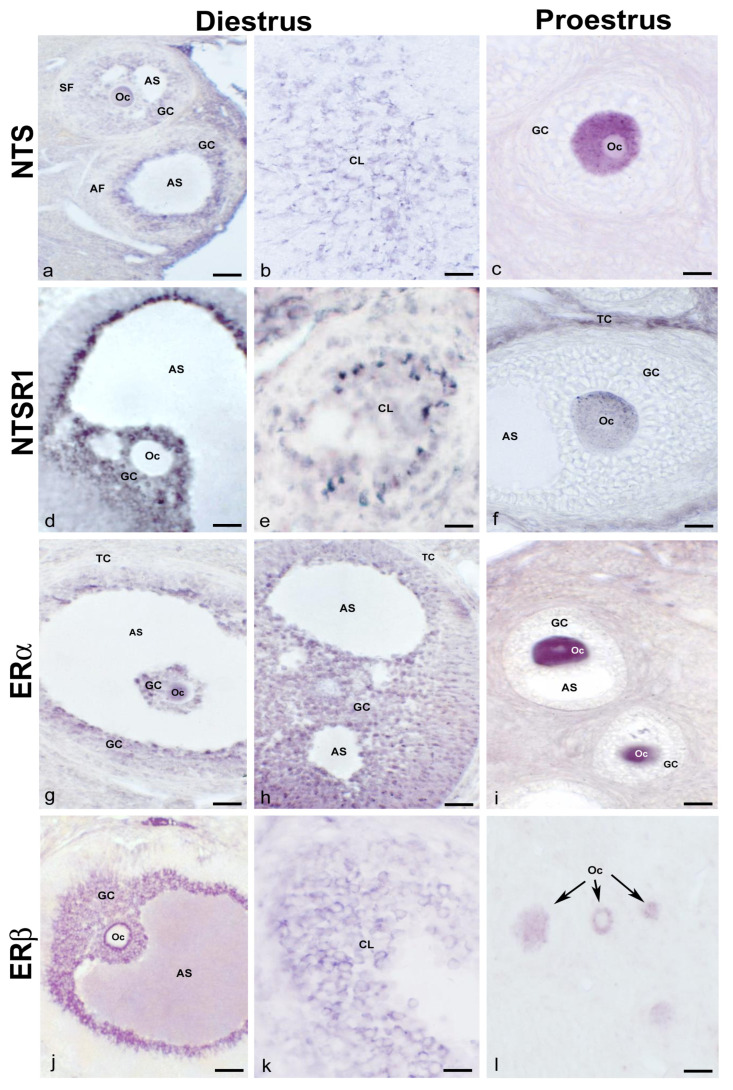
**NTS, NTSR1, and ERα/β in normal cycling adult females.** Representative images of the ovaries of cycling adult females showing NTS-ir in oocytes, granulosa cells from different follicles, the corpus luteum in diestrus phase (**a**,**b**), and in the oocyte cytoplasm of a multilayered primary follicle in proestrus phase (**c**); NTSR1-ir in granulosa cells of an antral follicle and cells of corpus luteum in diestrus phase (**d**,**e**), and the cytoplasm of the oocyte of a secondary follicle as well as in theca cells in proestrus phase (**f**); ERα-ir in the oocytes and granulosa cells of different follicles in the diestrus phase (**g**,**h**), and the cytoplasm and nucleus of the oocytes from secondary follicles in the proestrus phase (**i**); and ERβ-ir in the granulosa cells of an antral follicle and cells of the corpus luteum in the diestrus phase (**j**,**k**) and the cytoplasm of oocytes from primary follicles in the proestrus phase (arrows) (**l**). AF: antral follicle; AS: antral space; CL: corpus luteum; GC: granulosa cells; Oc: oocyte; SF: secondary follicle; TC: theca cells. Scale bars: (**a**,**b**): 50 µm; (**c**,**f**): 20 µm; (**d**,**g**,**h**,**j**): 60 µm; (**e**,**i**,**k**): 30 µm; and (**l**): 35 µm.

**Figure 2 ijms-24-07418-f002:**
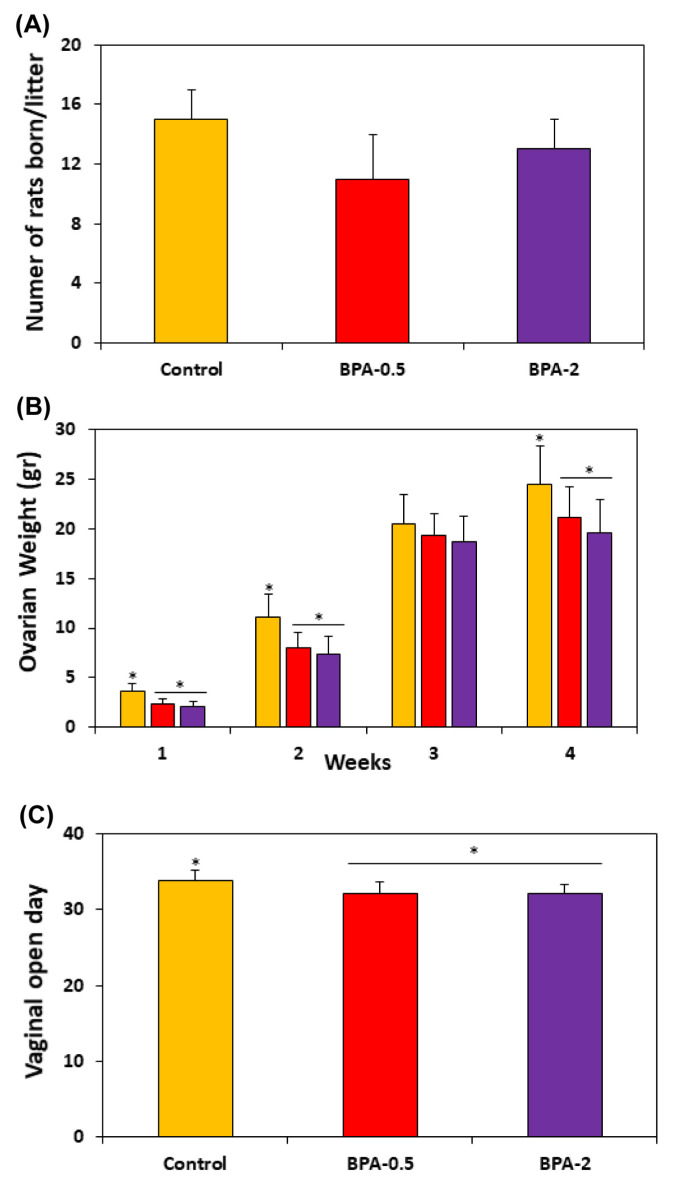
Graphs showing the number of rats born per litter (**A**); the ovarian weight in the offspring in the first four weeks of postnatal life (**B**); and the moment of the vaginal opening in the three experimental groups (**C**). Bars represent means ± SD (n = 5). The asterisks show significant differences between groups (*p* < 0.05).

**Figure 3 ijms-24-07418-f003:**
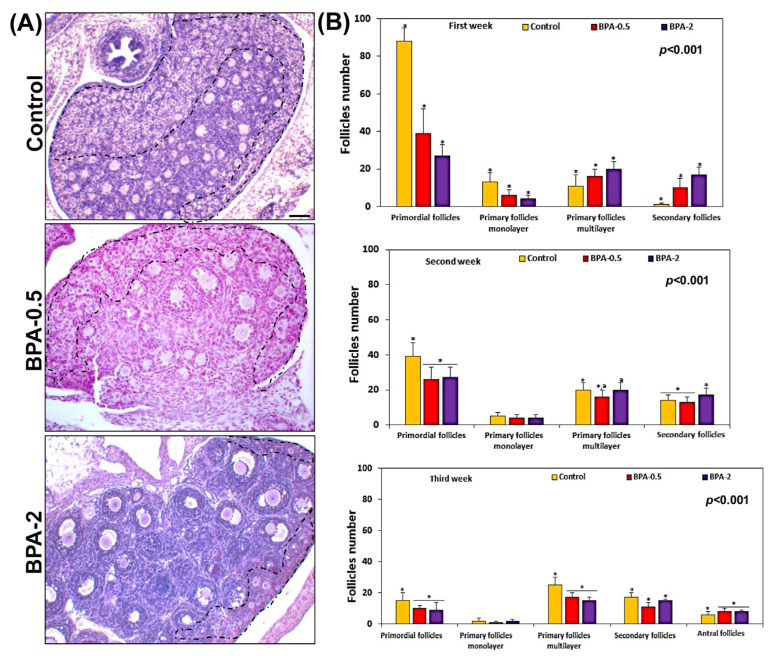
**Follicle quantification.** Representative images showing the density of follicles in the three experimental groups in the first week of postnatal life. The zone containing the primordial follicles has been roughly marked with the dashed line (**A**); graphs showing the number of each of the different types of ovarian follicles in the three experimental groups in the first three weeks of postnatal life (**B**). Bars represent means ± SD (n = 5). The identical symbol (asterisk) and letter on different bars indicates significant differences between groups. (**A**) Scale bar: 80 µm.

**Figure 4 ijms-24-07418-f004:**
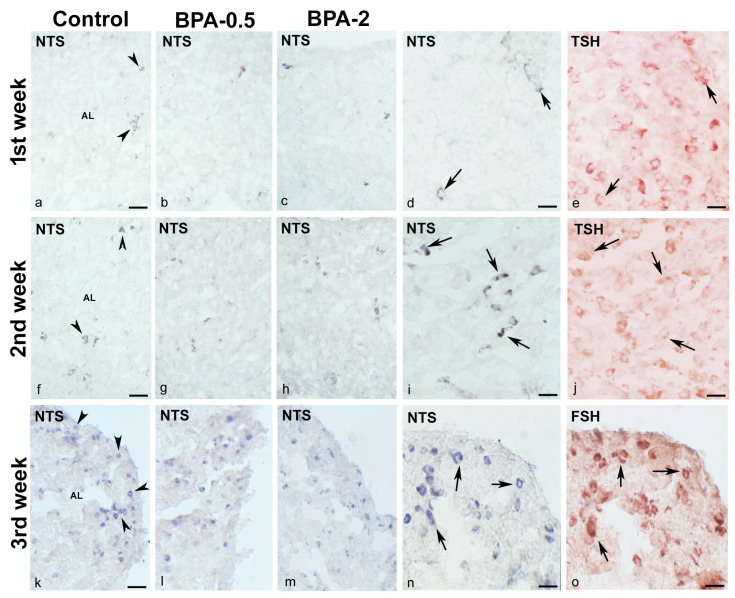
**NTS immunoreactivity in offspring pituitary glands.** Representative images of the pituitary gland in the first three weeks of postnatal life in the three experimental groups showing the presence of NTS-ir cells (arrowheads) in the central area of the anterior lobe in the 1st (**a**–**e**) and 2nd (**f**–**j**) weeks and in the margins of the anterior lobe in the 3rd week (**k**–**o**). Images (**d**,**e**) and (**i**,**j**) show NTS colocalization with TSH (arrows) in the 1st and 2nd weeks in control and BPA-0.5, respectively, and images (**n**,**o**) show NTS colocalization with FSH (arrows) in the 3rd week in BPA-2. AL: anterior lobe. Scale bars: (**a**–**c**), (**f**–**h**), and (**k**–**m**): 60 µm; (**d**,**e**), (**i**,**j**), and (**n**,**o**): 25 µm. (**a**–**d**), (**f**–**i**), and (**k**–**n**) immunoperoxidase chloronaphthol. (**e**,**j**,**o**) immunoperoxidase ethyl carbazole.

**Figure 5 ijms-24-07418-f005:**
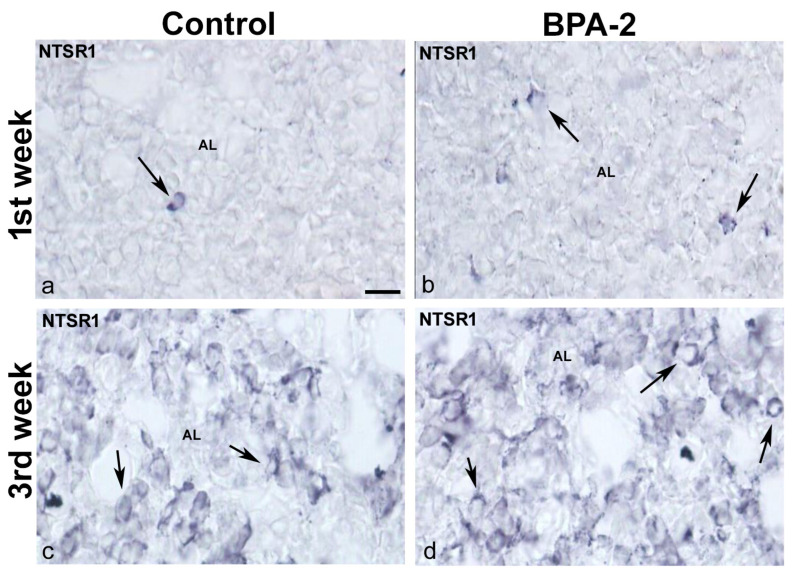
**NTSR1 immunoreactivity in the offspring pituitary gland.** Representative images of the pituitary gland in the first (**a**,**b**) and third (**c**,**d**) weeks of postnatal life showing the presence of NTSR1-ir cells (arrows) in the central area of the anterior lobe in the control and BPA-2 groups. AL: anterior lobe. Scale bar: 25 µm.

**Figure 6 ijms-24-07418-f006:**
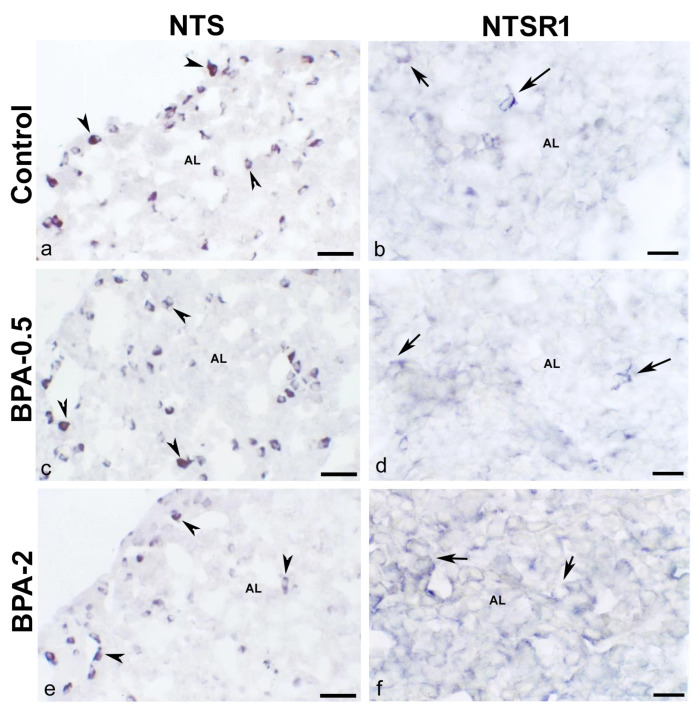
**NTS and NTSR1 immunoreactivity in the mother’s pituitary gland.** Representative images of the pituitary gland in mothers after weaning/lactation showing the presence of NTS-ir cells (arrowheads) (**a**,**c**,**e**) and NTSR1-ir cells (arrows) (**b**,**d**,**f**) in the three experimental groups. AL: anterior lobe. Scale bars: (**a**,**c**,**e**): 35 µm and (**b**,**d**,**f**): 25 µm.

**Figure 7 ijms-24-07418-f007:**
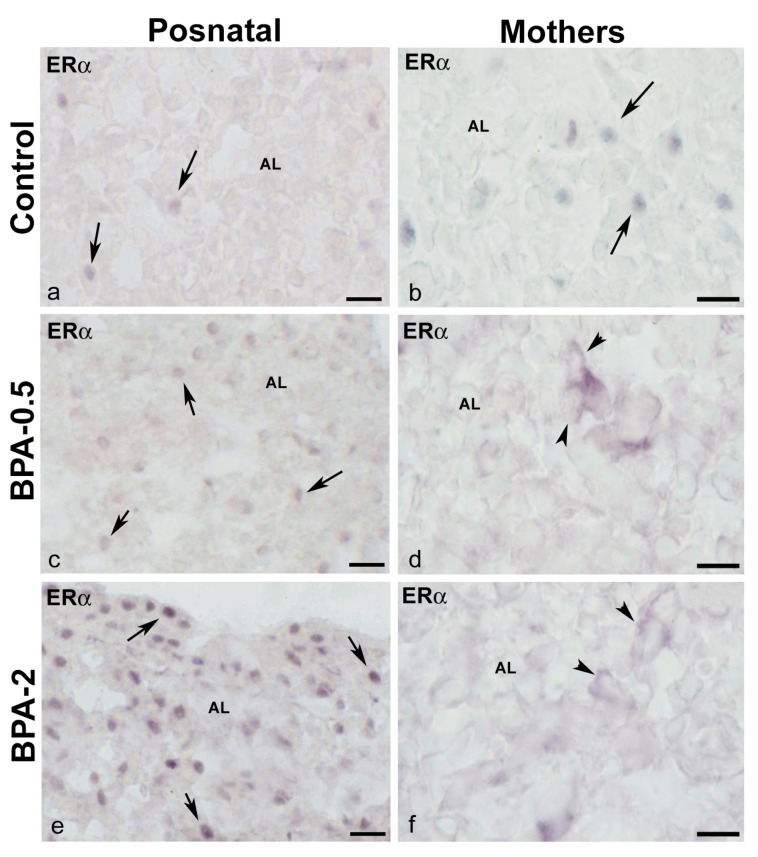
**ERα immunoreactivity in offspring and the mother’s pituitary gland**. Representative images of the pituitary gland in offspring at the first week of postnatal life (**a**,**c**,**e**) and mothers after weaning/lactation (**b**,**d**,**f**) showing the presence of ERα-ir cells in the anterior lobe in the three experimental groups. The nuclear location of ERα immunoreactivity in offspring and control mothers is indicated with arrows, while the cytosolic location in BPA-treated mothers is indicated with arrowheads. AL: anterior lobe. Scale bars: (**a**,**c**,**e**): 25 µm and (**b**,**d**,**f**): 20 µm.

**Figure 8 ijms-24-07418-f008:**
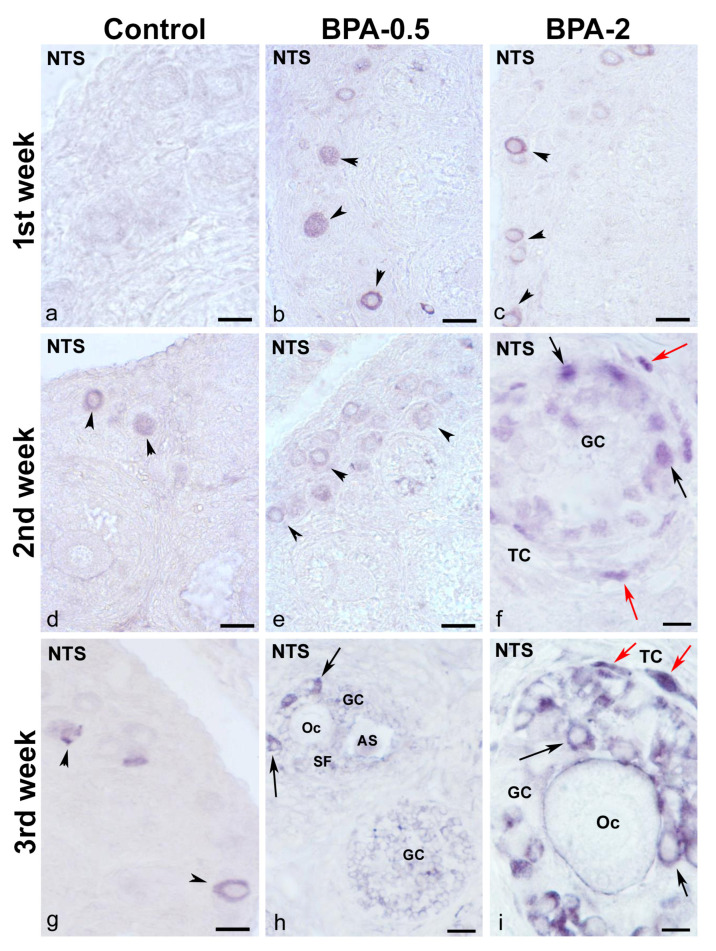
**NTS immunoreactivity in offspring ovaries**. Representative images of the ovary in the three experimental groups in the first three weeks of postnatal life showing the presence of NTS-ir in oocytes from primordial and primary follicles (arrowheads) (**a**–**e**,**g**) as well as in granulosa cells from multilayered primary (**f**,**i**) and secondary (**h**) follicles (black arrows) and in theca cells (red arrows). AS: antral space; GC: granulosa cells; Oc: oocyte; SF: secondary follicle; TC: theca cells. Scale bars: (**a**–**e**): 40 µm; (**f**): 12 µm; (**g**): 30 µm; (**h**): 15 µm; and (**i**): 10 µm.

**Figure 9 ijms-24-07418-f009:**
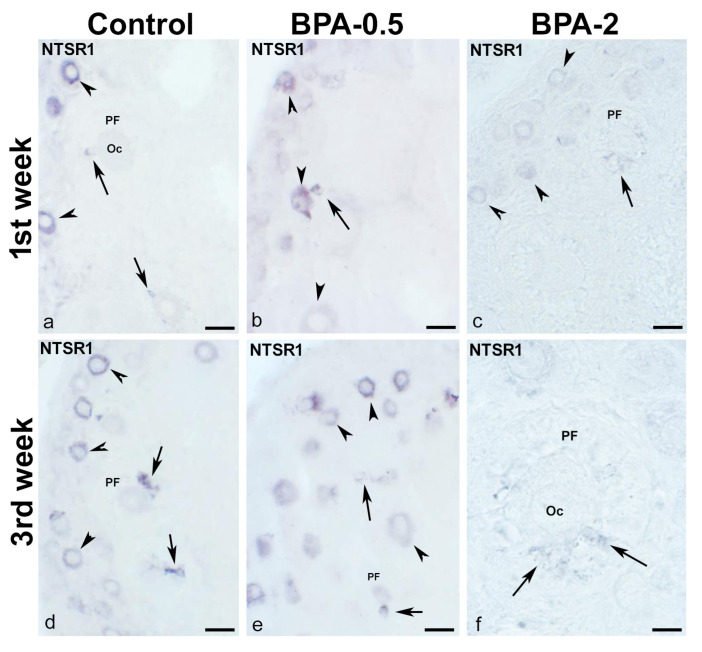
**NTSR1 immunoreactivity in offspring ovaries**. Representative images of the ovary in the three experimental groups in the first (**a**–**c**) and third (**d**–**f**) weeks of postnatal life show the presence of NTS-ir in oocytes from primordial follicles (arrowheads) as well as in granulosa cells from different primary follicles (arrows). Oc: oocyte; PF: primary follicle. Scale bars: (**a**–**e**): 40 µm; (**f**): 15 µm.

**Figure 10 ijms-24-07418-f010:**
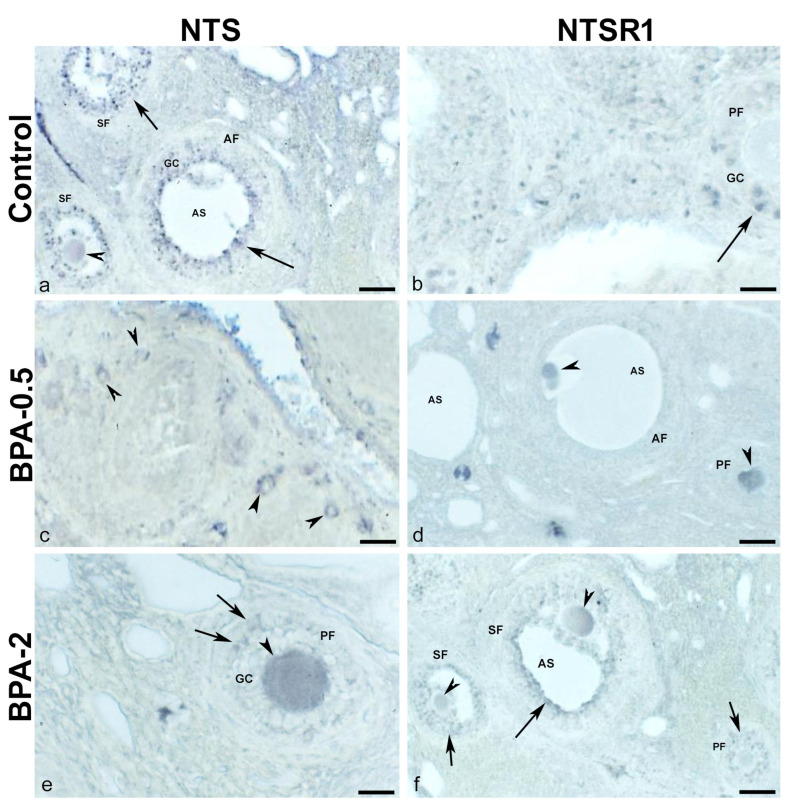
**NTS and NTSR1 immunoreactivity in the mother’s ovaries.** Representative images of the ovary in mothers after weaning/lactation showing the presence of NTS-ir (**a**,**c**,**e**) and NTSR1-ir (**b**,**d**,**f**) in oocytes (arrowheads) and granulosa cells (arrows) in the different follicles in the three experimental groups. AF: antral follicle; AS: antral space; GC: granulosa cells; PF: primary follicle; SF: secondary follicle. Scale bars: (**a**,**c**,**d**,**f**): 40 µm; (**b**,**e**): 20 µm.

**Figure 11 ijms-24-07418-f011:**
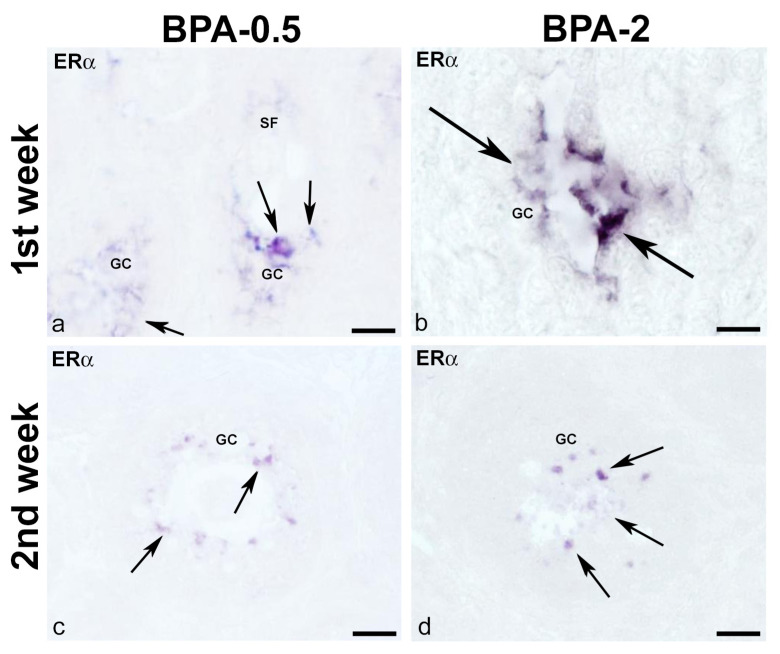
**ERα immunoreactivity in offspring ovaries.** Representative images of the ovary in the BPA-treated groups in the first (**a**,**b**) and second (**c,d**) weeks of postnatal life showing the presence of ERα-ir in granulosa cells (arrows) of secondary follicles. GC: granulosa cells; SF: secondary follicle. Scale bars: (**a**,**c**,**d**): 30 µm; (**b**) 15 µm.

**Figure 12 ijms-24-07418-f012:**
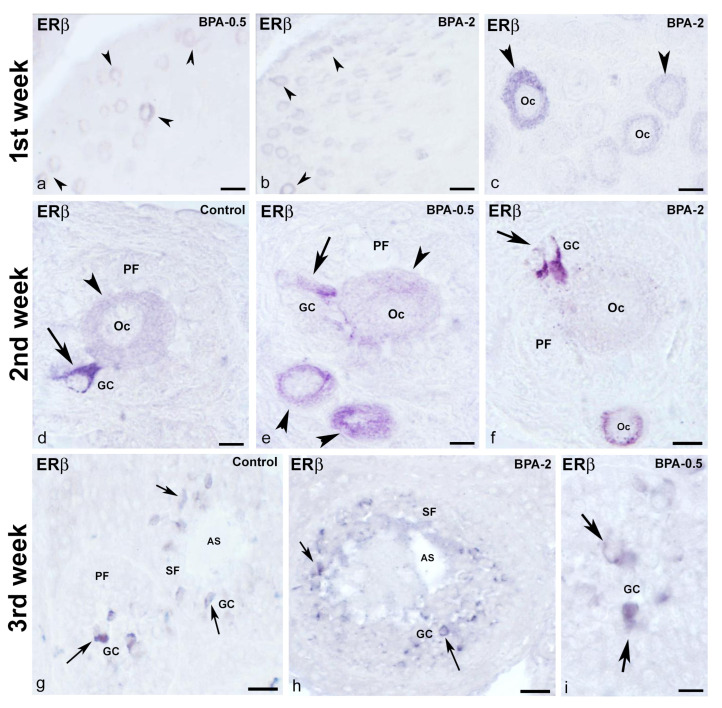
**ERβ immunoreactivity in offspring ovaries.** Representative images of the ovary in the three experimental groups in the three first weeks of postnatal life showing the presence of ERβ-ir in oocytes (arrowheads) from primordial follicles in the first week (**a**–**c**), in oocytes (arrowheads) from primordial and primary follicles as well as in granulosa cells (arrows) from primary follicles in the second week (**d**–**f**) and granulosa cells (arrows) from primary and secondary follicles in the third week (**g**–**i**). AS: antral space; GC: granulosa cells; Oc: oocyte; PF: primary follicle; SF: secondary follicle. Scale bars: (**a**,**b**): 40 µm; (**c**,**f**): 20 µm; (**d**,**e**,**i**): 10 µm; (**g**): 45 µm; (**h**): 36 µm.

**Figure 13 ijms-24-07418-f013:**
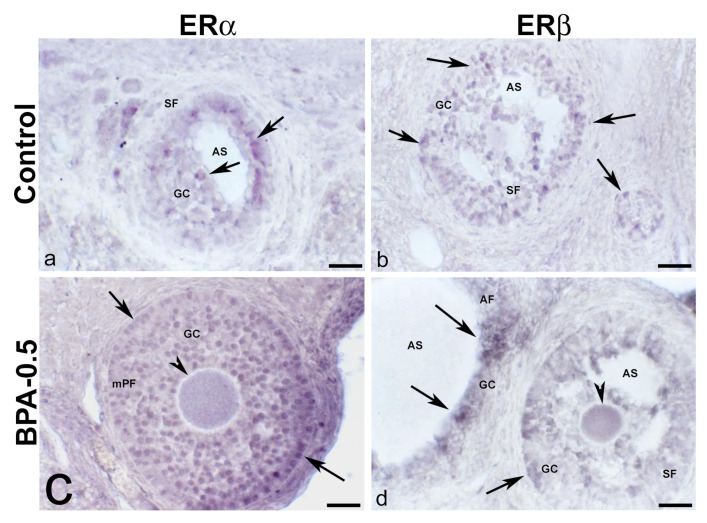
**ERα and ERβ immunoreactivity in the mother’s ovaries.** Representative images of the ovary in mothers after weaning/lactation, showing the presence of ERα-ir (**a**,**c**) and ERβ-ir (**b**,**d**) in oocytes (arrowheads) and granulosa cells (arrows) from multi-layered primary follicle, secondary and antral follicles in the control and BPA-0.5 groups. AF: antral follicle; AS: antral space; GC: granulosa cells; mPF: multilayered primary follicle; SF: secondary follicle. Scale bars: (**a**–**d**): 60 µm.

## Data Availability

The data used to support the findings of this study are available from the corresponding author upon request.

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
