# Peer review of "NTS, NTSR1 and ERs in the Pituitary–Gonadal Axis of Cycling and Postnatal Female Rats after BPA Treatment"

_ijms, 2023, doi:10.3390/ijms24087418_

Round 1

Reviewer 1 Report

The manuscript fits the scope of the journal and the overall research is well done. However, a minor but critical detail is missing, without which the manuscript should not be considered for publication. The manuscript lacks the Institutional Review Board Statement, i.e. the animal study protocol approved by the Institutional Review Board (or Ethics Committee); a protocol code number and approval date must be added.

Also, as far as the manuscript concerning, figures 10 and 13 should be improved. Images in figure 10 and 13 appear blurred and the immunoreactivity is not particularly evident. in figure 10,  higher magnification may help recognize the immunoreactive signals. 

Author Response

Reviewer 1

The manuscript fits the scope of the journal and the overall research is well done. However, a minor but critical detail is missing, without which the manuscript should not be considered for publication. The manuscript lacks the Institutional Review Board Statement, i.e. the animal study protocol approved by the Institutional Review Board (or Ethics Committee); a protocol code number and approval date must be added.

We appreciate the reviewer's comments and we have obtained the mandatory favorable report from the Ethics Committee for animal Care of the University of La Laguna (CEIBA) with the identification code CEIBA2023-3289. The report  information with the identification code number and approval date have been added to the manuscript. The original report from the Ethics Committee is sent as an attachment (author-coverletter-28177130.v1.).

Also, as far as the manuscript concerning, figures 10 and 13 should be improved. Images in figure 10 and 13 appear blurred and the immunoreactivity is not particularly evident. in figure 10, higher magnification may help recognize the immunoreactive signals. 

We appreciate the reviewer's suggestions and following the recommendations, the images corresponding to figures 10 and 13 have been improved and enlarged in the case of figure 10 so that in the revised manuscript, the results in both figures can be observed more clearly.

Reviewer 2 Report

The authors performed a very interesting experimental study investigating the effect of bisphenol a on NTS, NTSR1 and ERs in the pituitary-gonadal axis. After exposing female pregnant rats to BPA at 0.5 and 2 mg/kg per day during gestation and lactation was monitored by indirect immunohistochemistry. 

The study is very interesting as it proves the direct association of BPA with the induction of morphohistological changes in the gonads of female offspring rats. 

The only issue that i would like the authors to comment is the actual relevance of the used dosage to standard BPA exposure in real life and if their study provides evidence for population with standard environmental or occupation exposure, as their study should provide direct information for clinical practice and physicians. 

Author Response

Reviewer 2

The authors performed a very interesting experimental study investigating the effect of bisphenol a on NTS, NTSR1 and ERs in the pituitary-gonadal axis. After exposing female pregnant rats to BPA at 0.5 and 2 mg/kg per day during gestation and lactation was monitored by indirect immunohistochemistry. 

The study is very interesting as it proves the direct association of BPA with the induction of morphohistological changes in the gonads of female offspring rats. 

The only issue that I would like the authors to comment is the actual relevance of the used dosage to standard BPA exposure in real life and if their study provides evidence for population with standard environmental or occupation exposure, as their study should provide direct information for clinical practice and physicians. 

The dose of BPA used in this work approximates the real doses to which the population is exposed. The real dose is high since BPA is present in many products for daily use, but it is also variable taking its hormonal behavior, so neither the dose nor the effects can be extrapolated to the human population. On the one hand, high doses can have the opposite effect to low ones and, on the other hand, each person can be exposed to different doses and, in addition, their physical characteristics, physiological state, stage of life and gender will influence the body's response to BPA.

In this study, we provide an example of how this molecule can affect processes as important as the regulation of hormone production and oogenesis.

This work has an important part of basic research where we can observe in the pituitary and ovarian cells the expression of factors that have functions in the gonadal axis and therefore allow us to predict what their effects on reproductive life would be.

In this study we have shown that BPA acts like an estrogen, so we might think that exposure to BPA would imply an increase in estrogenic effects in humans, especially at times such as development where the levels of sex hormones are very important in processes as the development of the central nervous system

Previous findings from our group have shown that in humans, as well as in rats, NTS is involved in the function of the same pituitary cells, and therefore it can be assumed that it is involved in the gonadal axis.

Certain findings of this work, such as the decrease in the number of primordial follicles, are a very important fact to take into account in the duration of fertile life in humans.

In conclusion, we believe that there is no direct application of our results in the clinic, but they provide more information on how this molecule can affect fertility and reproductive function.